# Reliability of Synthetic Brain MRI for Assessment of Ischemic Stroke with Phantom Validation of a Relaxation Time Determination Method

**DOI:** 10.3390/jcm9061857

**Published:** 2020-06-14

**Authors:** Chia-Wei Li, Ai-Ling Hsu, Chi-Wen C. Huang, Shih-Hung Yang, Chien-Yuan Lin, Charng-Chyi Shieh, Wing P. Chan

**Affiliations:** 1Department of Radiology, Wan Fang Hospital, Taipei Medical University, Taipei 116, Taiwan; chiawei.lee@gmail.com (C.-W.L.); irene751110@gmail.com (A.-L.H.); cth.mail05@gmail.com (C.-W.C.H.); schpeltor_tw@yahoo.com.tw (S.-H.Y.); 2Department of Radiology, School of Medicine, College of Medicine, Taipei Medical University, Taipei 110, Taiwan; 3GE Healthcare, Taipei 104, Taiwan; Eddy.Lin@ge.com (C.-Y.L.); mark.shieh@ge.com (C.-C.S.); 4Medical Innovation Development Center, Wan Fang Hospital, Taipei Medical University, Taipei 116, Taiwan

**Keywords:** brain, magnetic resonance imaging, technology, white matter

## Abstract

The reliability of relaxation time measures in synthetic magnetic resonance images (MRIs) of homemade phantoms were validated, and the diagnostic suitability of synthetic imaging was compared to that of conventional MRIs for detecting ischemic lesions. Phantoms filled with aqueous cupric-sulfate (CuSO_4_) were designed to mimic spin-lattice (T_1_) and spin-spin (T_2_) relaxation properties and were used to compare their accuracies and stabilities between synthetic and conventional scans of various brain tissues. To validate the accuracy of synthetic imaging in ischemic stroke diagnoses, the synthetic and clinical scans of 18 patients with ischemic stroke were compared, and the quantitative contrast-to-noise ratios (CNRs) were measured, using the Friedman test to determine significance in differences. Results using the phantoms showed no significant differences in the interday and intersession synthetic quantitative T_1_ and T_2_ values. However, between synthetic and referenced T_1_ and T_2_ values, differences were larger for longer relaxation times, showing that image intensities in synthetic scans are relatively inaccurate in the cerebrospinal fluid (CSF). Similarly, CNRs in CSF regions of stroke patients were significantly different on synthetic T_2_-weighted and T_2_-fluid-attenuated inversion recovery images. In contrast, differences in stroke lesions were insignificant between the two. Therefore, interday and intersession synthetic T_1_ and T_2_ values are highly reliable, and discrepancies in synthetic T_1_ and T_2_ relaxation times and image contrasts in CSF regions do not affect stroke lesion diagnoses. Additionally, quantitative relaxation times from synthetic images allow better estimations of ischemic stroke onset time, consequently increasing confidence in synthetic MRIs as diagnostic tools for ischemic stroke.

## 1. Introduction

In synthetic magnetic resonance images (MRIs), multiple sets of sequence properties are used to obtain quantitative values for proton density, longitudinal relaxation time (T_1_), and transverse relaxation time (T_2_) at each imaging voxel. By adjusting the imaging parameters repetition time (TR), echo time (TE), and inversion time (TI), a technologist can mathematically produce multiple image contrast weightings from a single synthetic scan [1,2,3] once the quantitative values for T_1_, T_2_, and proton density are generated. Synthetic MRI technology using a two-dimensional fast spin-echo (SE) multi-dynamic, multi-echo sequence was recently approved by the US Food and Drug Administration and has also been used in clinical brain studies [4]. Synthetic MRIs allow completion of diagnostic brain examinations in about 4 to 7 min using a single scan. The shorter scan time is beneficial to stroke patients because it increases the diagnostic procedure’s efficiency and comfort. Synthetic MRIs also provide a distinct tissue relaxation time [5,6,7], enabling a more accurate estimation of ischemic stroke status [8,9]. Because establishing an optimum relaxation time is a prerequisite for generating a synthetic image, achieving favorable synthetic image contrast depends on the accuracy of that value.

This study was conducted to validate the reliabilities of T_1_ and T_2_ relaxation times obtained from synthetic MRIs, and to investigate the achievability of sufficient contrast between stroke lesions and normal brain tissues using synthetic MRIs.

## 2. Materials and Methods

Imaging was performed using a 3T clinical scanner (MR750w; GE Healthcare, Milwaukee, WI, USA) with a body coil as the transmitter and a 24-channel array head coil as the receiver. The research protocol was approved by the X Medical University Joint Institutional Review Board (No. N201706024). All participants gave written informed consent.

### 2.1. Synthetic MRI Technique

Synthetic scans were obtained using MRI compilation (MAGiC) software and one multi-dynamic multi-echo sequence with the following parameters: repetition time (TR), 4123 ms; TE, 21.4 and 96.3 ms; and echo-train length, 14 echoes. Shared parameters across all scans were: matrix size, 256 × 256; slice thickness, 5 mm; slice gap, 1 mm; frequency, 20 slices per scan; and field of view, 160 × 160 mm for phantom scans and 192 × 192 mm for brain scans. Scan time was about 4 min.

### 2.2. Specifically Designed Phantoms

This study was conducted to validate the reliabilities of T_1_ and T_2_ relaxation times from synthetic MRIs of phantoms containing sealed tubes filled with specific concentrations of aqueous CuSO_4_ (anhydrous copper sulfate; 97.5% purity; Nacalai Tesque, Kyoto, Japan) and fixed in a plastic tank (Figure 1). Two phantoms were designed specifically to give T_1_ and T_2_ values between about 1100 and 2300 ms and 50 and 240 ms, respectively [10], mimicking the primary components of the brain: white matter (WM), gray matter (GM), and cerebrospinal fluid (CSF) [11,12,13,14].

### 2.3. Phantom Evaluation

The reference T_1_ value was quantified using a conventional SE with an inversion recovery sequence having the following parameters: TR, 10,000 ms; TE, 8.9 ms; and a TI series (100, 300, 500, 700, 800, 900, 1000, 1100, 1200, 1300, 1400, 1500, 1800, 2400, 3000, and 4000 ms). The reference T_2_ value was quantified using a multi-echo fast-SE sequence with the following parameters: TR, 10,000 ms, and a TE series (9.9, 19.9, 29.8, 39.7, 49.6, 59.6, 69.5, 79.4, 89.4, 99.3, 109.2, 119.2, 129, 139, 149, 158.8, 168.7, 178.6, 188.5, and 198.4 ms). Both values from all scans were calculated using an in-house MATLAB script for nonlinear least-squares curve fitting (MathWorks, Natick, MA, USA) [15]. Interday reproducibility was assessed using the imaging coverage applied in conventional scans. Data were acquired from both phantoms on two days, one week apart, performing seven repetitions at 30-min intervals, and all synthetic data were used to assess intersession repeatability. The percent difference between the synthetic and conventional values was calculated as:Difference = ((T_synthetic scan_ − T_reference_)/T_reference_) × 100%.

The differences obtained using the concentrations of aqueous CuSO_4_ in those two specific phantoms were calculated for quantitative validation.

### 2.4. Participant and Diagnostic Score Evaluation

Twenty patients with chronic stroke (>6 months), free of known neurologic and psychiatric diagnoses, were enrolled between 20 August 2017, and 28 December 2017, to explore the image contrasts in synthetic scans and to quantitatively compare synthetic and clinical images. Two were removed from the analysis due to excessive head movement. The mean (standard deviation, SD) of the resulting sample of 18 patients (14 men) was 59.6 (6.4) years.

Standardized stroke MRIs and synthetic scans were evaluated using double-blind testing. Two radiologists rated image quality on an 11-point scale (0, entirely poor to 10, entirely good) and answered one question: “Are these images acceptable for diagnostic use? (yes or no).” The display order of the patient images was randomly generated by MATLAB.

Images were prospectively acquired using a fixed set of parameters closely approximating a standardized stroke MRI protocol [16]. Images of clinical stroke were acquired first; then synthetic scans were obtained, including fast-SE with inversion-recovery axial plane T_1_-weighted imaging (TR, 1800 ms; TE, 23 ms; TI, 750 ms), fast-SE coronal plane T_2_-weighted imaging (TR, 4545 ms; TE, 110 ms), and T_2_-fluid-attenuated inversion recovery (FLAIR) imaging (TR, 9000 ms; TE, 92 ms; TI, 2472 ms). One echo-planar-imaging-based diffusion-weighted image (TR, 6000 ms; TE, 77 ms) was also acquired, using 1000 s/mm² for the b-value, and one experienced radiologist located the ischemic lesion in each patient (Table 1).

Preprocessing, analysis, and spatial segmentation were applied to all datasets, and mean image intensities for WM, GM, and CSF were extracted. The ICBM-152 [17,18] and Anatomical Automatic Labeling templates [19,20] were normalized into individual images of each patient, and the mean signal intensity of the bilateral thalamus was extracted for image contrast standardization (Figure 2). The standardized image contrast-to-noise ratio (CNR) was calculated as:CNR = (mean intensity of WM/GM/CSF/stroke lesion - mean intensity of thalamus)/standard deviation of thalamus.

### 2.5. Statistical Analyses

Statistical analyses were performed using the R software program with *t* test functionality, version 3.5.1 (R Core Team 2018. R: A language and environment for statistical computing. R Foundation for Statistical Computing, Vienna, Austria. URL http://www.R-project.org/). The Friedman test was used to measure interday and intersession repeatabilities and the image contrast between clinical and synthetic images, allowing quantitative validation of phantom and patient data. A difference was considered statistically significant when the Bonferroni-corrected *p*-value was <0.05.

## 3. Results

Patient demographic data and the clinical locations of the stroke lesions are shown in Table 1.

### 3.1. Phantom Evaluation

Table 2 shows the interday and intersession quantitative T_1_ and T_2_ values across all applicable concentrations of aqueous CuSO_4_, acquired using synthetic scanning. The Friedman test showed no significant differences between the interday and intersession values. The reference T_1_ and T_2_ values were obtained using conventional SE methods. Differences in T_1_ and T_2_ between the synthetic and conventional scans are shown in Figure 3. Synthetic T_1_ values were closer to the reference values when CuSO_4_ concentrations were higher (at 0.7 to 1.0 mmol/L, the difference was 5–8%, whereas at <0.2 mmol/L, the difference was >20%). A similar trend was seen for synthetic T_2_ values (at 11.0 to 20.0 mmol/L, the difference was <10%, whereas at 7.0 to 9.0 mmol/L, the difference was >20%).

### 3.2. Participant Evaluation

Table 3 shows the subjective ratings by two radiologists of the quality of the synthetic and clinical images and the suitability of synthetic images for stroke diagnosis. The clinical and synthetic images were identically rated for diagnostic suitability. However, the image quality rating, which exceeded 8 for synthetic T_1_- and T_2_-weighted scans and exceeded 7.8 for T_2_-FLAIR synthetic scans, was significantly greater for the corresponding clinical scans (Bonferroni-corrected *p* = 0.05). Figure 4 shows representative clinical and synthetic scans of a stroke patient, demonstrating that the contrasts are similar. Instead of visually evaluating the image contrast, the CNR was determined in WM, GM, CSF, and stroke lesions (Figure 5). Differences were not significant in any of the WM and GM regions (see Figure 5) but were significant in the CSF regions on T_2_-weighted and T_2_-FLAIR images (Bonferroni-corrected *p* = 0.05, multiple-comparison correction). In the lesion regions, CNRs were not significantly different between images reconstructed from synthetic scans and those acquired using clinical scanning methods.

## 4. Discussion

In this study, significant differences in T_1_ and T_2_ values across the 14 synthetic scans of the two phantoms were not found; therefore, the reliability of acquiring synthetic quantitative T_1_ and T_2_ values is high. The relaxation times varied across brain tissues, but smaller differences were found between the synthetic and reference values when they ranged from about 1131 to 1456 ms for T_1_ and from about 59 to 94 ms for T_2_. Differences were apparent in the CSF when T_1_ and T_2_ were greater than 1456 ms and 94 ms, respectively. As seen with the phantoms, differences were large when relaxation times were in the range for CSF regions, confirming that results in CSF regions account for the differences found. Significant differences were not found between synthetic and clinical T_2_-FLAIR images of stroke lesions, showing that disparities in signal intensity from the various CSF regions did not affect the diagnosis of chronic stroke lesion, most commonly located in WM and GM. The ratings by two radiologists for diagnostic suitability concur with this result.

The literature demonstrates that the quantitative T_2_ values of stoke lesions are significantly correlated with time since symptom onset [9]. Quantitative T_2_ values are significantly higher for ischemic lesions than for healthy tissue, and using T_2_ values to predict time since symptom onset is more accurate than using FLAIR images. One animal study [8] showed that quantitative values for T_1_ and T_2_ increased linearly with ischemia duration; in addition, a more accurate estimation of stroke duration was achieved using a pixel-by-pixel analysis to quantify relaxation times at a single point in time than by using signal intensity. Moreover, quantitative relaxation times provide additional information about tissue status in ischemia. Uncertainties in onset time estimates using quantitative T_1_ and T_2_ values vary from about 25 to 47 min in the first 5 h of ischemic stroke [21]. Therefore, in clinical images, estimating the onset time and identifying ischemic tissue are more accurate using quantitative relaxation times rather than signal intensities; they are free from operator bias, thereby increasing the reproducibility of time since symptom onset. However, they might not have a clinical application in diagnosing ischemic stroke because dozens of minutes are always required to quantify relaxation times using traditional clinical scanning methods. The synthetic technique uses the multi-dynamic, multi-echo sequence with four fixed inversion times and two echo times to simultaneously acquire quantitative values (including T_1_, T_2_, and proton density) within a couple minutes. This efficiency allows quantitative relaxation times to be used for ischemia diagnoses.

Our results show no significant differences in stroke lesion regions between images reconstructed from synthetic T_2_-FLAIR scans and those acquired using clinical scans. However, the image quality ratings of the synthetic T_2_-FLAIR scans were significantly lower than those for the corresponding clinical scans due to several artifacts in the former. These included granular hyperintensity signals occasionally appearing in the margins and flow artifacts [4,6], resulting from partial volume effects and flow effects not considered in the analytical signal model when estimating image parameters. Artifacts in synthetic T_2_-FLAIR scans resulted in lower image quality but showed little interference in diagnosing chronic stroke. Significant differences were also found between the synthetic and clinical T_2_-FLAIR images in the CSF regions; two radiologists found that the synthetic images were of lower quality, agreeing with the findings of others [4]. Two recent studies used deep learning algorithms to solve this problem. Hagiwara et al. attempted to improve the accuracy in brain CSF regions by using conditional generative adversarial network training to perform pixel-by-pixel translation methods, thus generating FLAIR images with contrasts closer to those of conventional FLAIR images [22]. The corrected synthetic FLAIR images provided Dice scores and image qualities that were closer to or more comparable with those found in conventional FLAIR images, and they showed fewer granular and swelling artifacts. Ryu et al. used 56 FLAIR datasets to train a deep-learning-based synthetic FLAIR method to correct for artifacts in synthetic FLAIR images [23], significantly improving the normalized root-mean-square values and the structural similarities. The results of these two studies suggest possible methods for correcting artifacts in synthetic images.

Our study has two limitations: the simple structure of our phantoms and the small sample size. First, in this study, we designed and used two phantoms with 14 specific concentrations of aqueous CuSO4, showing that the T_1_- and T_2_-values are similar to those of brain tissue, thus validating the accuracy of quantitative relaxation times acquired by synthetic scanning. However, brain tissues are anisotropic structures and can affect relaxation processes, and the simple structure of our phantoms could not describe actual brain tissue but only changes in free fluid spaces. Some have demonstrated that other types of phantoms could better reflect the reality in brain tissue compared to ours [24,25,26,27,28]; therefore, we suggest using those phantoms in further studies validating quantitative relaxation times. Second, this infers that our results are not generalizable to other comparisons between synthetic and clinical scanning in chronic ischemic stroke. Therefore, these results cannot be applied to acute/sub-acute stroke or hemorrhagic stroke. Further investigations using images in acute, sub-acute, and hemorrhagic stroke are now necessary.

In summary, this study shows high interday and intersession reliabilities of quantitative relaxation times in synthetic scans. Although discrepancies in T_1_ and T_2_ relaxation times and image contrasts in CSF regions were observed in synthetic scans, they did not affect the diagnoses of stroke lesions. Also, quantitative measures of relaxation times by such a fast and efficient means can improve the accuracy of ischemic stroke assessment, consequently increasing confidence in applying synthetic MRIs to ischemic stroke diagnoses.

## Figures and Tables

**Figure 1 jcm-09-01857-f001:**
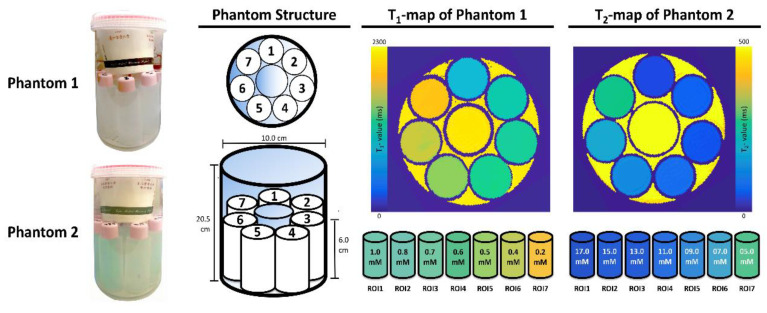
The structures of two phantoms designed for validating T_1_- and T_2_-value quantitation. Plastic tubes (cylinder structure with 7.1 cm^2^ in dimension and 6.0 cm in height) filled with various concentrations of CuSO_4_ were sealed and fixed in the plastic tanks. The plastic tank was filled with normal saline to minimize image artifacts.

**Figure 2 jcm-09-01857-f002:**
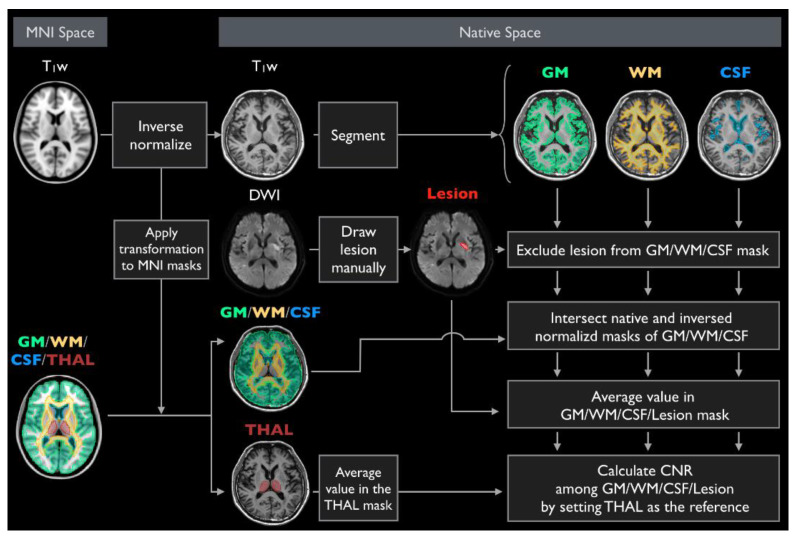
Data processing flowchart for quantitative tissue contrast validating synthetic scans.

**Figure 3 jcm-09-01857-f003:**
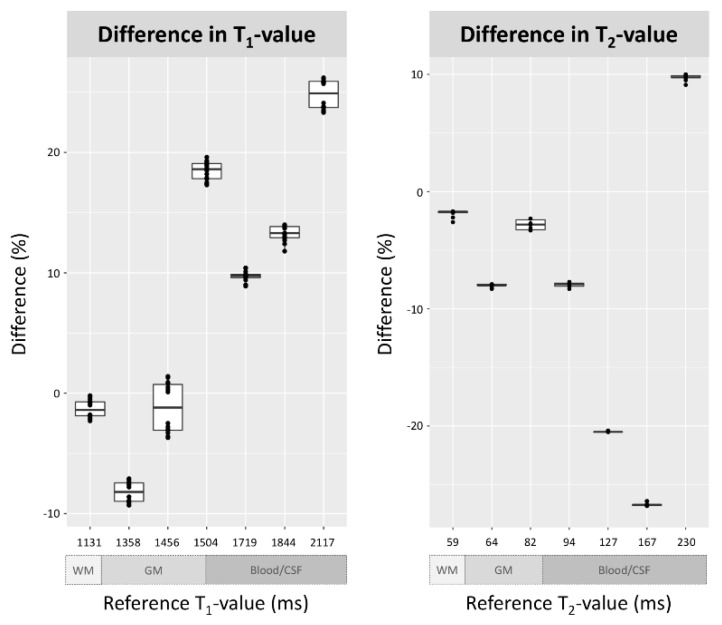
The difference in T_1_- and T_2_-values of two aqueous CuSO_4_ phantoms between the synthetic and clinical scans.

**Figure 4 jcm-09-01857-f004:**
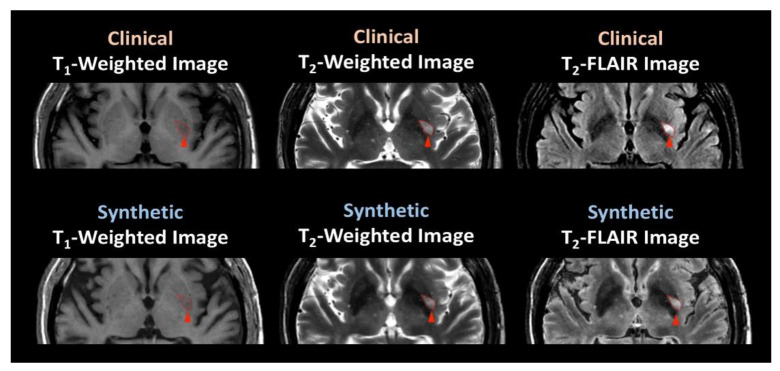
Images acquired by clinical and synthetic scanning of a 64-year-old male patient with stroke.

**Figure 5 jcm-09-01857-f005:**
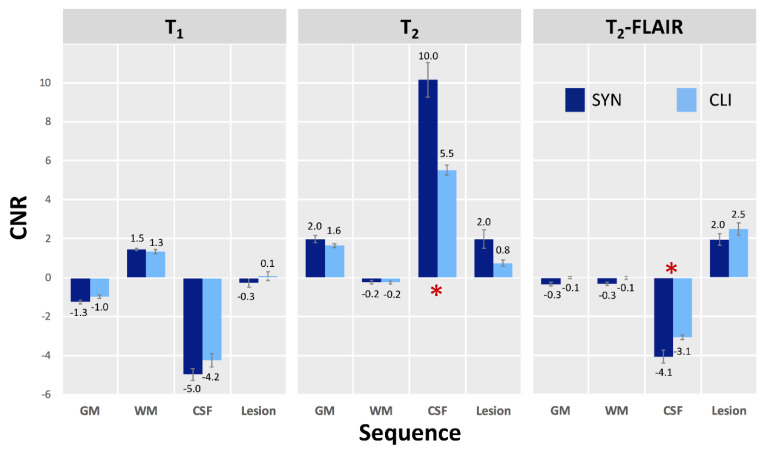
The standardized contrast-to-noise ratio (CNR) in synthetic and clinical scans of four brain regions. Significant differences were found between synthetic and clinical T_2_-weighted and T_2_-FLAIR images of cerebrospinal fluid regions (* Bonferroni corrected *p* < 0.05).

**Table 1 jcm-09-01857-t001:** Demographic data of each included stroke patient.

Patient (*n* = 18)	Age (Year)	Sex	Lesion Location(s)
Patient 01	64	Male	Left striatum
Patient 02	44	Female	Left striatum
Patient 03	64	Male	Left cerebellum
Patient 04	66	Male	Left striatum
Patient 05	63	Female	Right temporal and frontal lobes
Patient 06	50	Male	Left parietal lobe
Patient 07	66	Male	Left frontal lobe
Patient 08	55	Male	Left striatum
Patient 09	56	Male	Left parietal lobe
Patient 10	56	Female	Left insula
Patient 11	58	Female	Right parietal lobe
Patient 12	69	Male	Left striatum
Patient 13	55	Male	Right striatum
Patient 14	62	Male	Left cerebellum
Patient 15	57	Male	Left parietal lobe and right striatum
Patient 16	57	Male	Left striatum
Patient 17	66	Male	Right striatum
Patient 18	64	Male	Right parietal lobe

**Table 2 jcm-09-01857-t002:** Average quantitative values acquired using conventional magnetic resonance imaging (MRI) spin-echo sequences and synthetic MRI scans.

**T_1_ Value**															
**CuSO_4_ Concentration**	**Reference Value**	**Quantitative Value, Synthetic Scan, Day 1**							**Quantitative Value, Synthetic Scan, Day 2**						
		**1–1**	**1–2**	**1–3**	**1–4**	**1–5**	**1–6**	**1–7**	**2–1**	**2–2**	**2–3**	**2–4**	**2–5**	**2–6**	**2–7**
**1.0 mM**	1131.5	1123.1	1128.8	1122.6	1127.7	1122.2	1119.9	1126.2	1110.5	1111.0	1111.1	1105.6	1107.5	1111.2	1107.8
**0.8 mM**	1357.8	1254.1	1254.9	1257.5	1260.1	1252.1	1262.0	1257.9	1240.8	1241.3	1231.0	1231.7	1236.6	1235.1	1233.9
**0.7 mM**	1456.4	1463.0	1474.8	1476.3	1470.0	1460.8	1468.6	1457.3	1419.8	1416.0	1411.2	1402.4	1412.8	1407.9	1404.4
**0.6 mM**	1503.7	1785.4	1798.6	1794.5	1790.5	1787.7	1781.8	1793.0	1789.3	1776.9	1767.5	1764.9	1771.9	1772.5	1764.5
**0.5 mM**	1719.0	1883.3	1892.6	1889.5	1885.2	1879.7	1874.3	1898.2	1887.6	1884.6	1872.3	1887.2	1897.6	1888.2	1885.8
**0.4 mM**	1843.8	2071.6	2088.3	2086.3	2082.1	2077.3	2061.3	2084.3	2099.5	2095.6	2088.2	2096.7	2101.7	2099.8	2100.5
**0.2 mM**	2117.2	2612.3	2621.5	2620.8	2620.0	2614.7	2610.0	2628.2	2663.2	2665.0	2662.4	2665.8	2669.1	2666.3	2671.7
**T_2_ Value**															
**CuSO_4_ Concentration**	**Reference Value**	**Quantitative Value, Synthetic Scan, Day 1**							**Quantitative Value, Synthetic Scan, Day 2**						
		**1–1**	**1–2**	**1–3**	**1–4**	**1–5**	**1–6**	**1–7**	**2–1**	**2–2**	**2–3**	**2–4**	**2–5**	**2–6**	**2–7**
**20.0 mM**	58.5	57.4	57.5	57.5	57.5	57.4	57.4	56.9	57.4	57.5	57.5	57.5	57.4	57.4	57.2
**15.0 mM**	64.0	58.9	58.9	58.9	58.9	58.9	58.9	58.7	58.9	58.9	58.9	58.9	58.9	58.9	58.7
**13.0 mM**	82.4	80.5	80.6	80.1	80.1	79.7	79.7	80.2	80.5	80.6	80.1	80.1	79.7	79.7	79.9
**11.0 mM**	94.0	86.4	86.6	86.6	86.2	86.7	86.7	86.8	86.4	86.6	86.6	86.2	86.7	86.7	86.5
**9.0 mM**	126.7	100.8	100.7	100.7	100.7	100.8	100.8	100.8	100.8	100.7	100.7	100.7	100.8	100.8	100.7
**7.0 mM**	166.9	122.9	122.4	122.4	122.4	122.2	122.2	122.3	122.9	122.4	122.4	122.4	122.2	122.2	122.4
**5.0 mM**	230.3	253.3	252.7	253.2	252.2	252.8	252.8	253.1	253.2	252.5	253.1	252.1	252.7	252.7	251.3

**Table 3 jcm-09-01857-t003:** Diagnostic image quality rated by two radiologists.

Question	T1-Weighted Image		T2-Weighted Image		T2-FLAIR Image	
	SYN	CLI	SYN	CLI	SYN	CLI
What is the image quality for diagnostic use? ^a^	8.06 ** (0.79)	9.42 (0.28)	8.53 * (1.00)	9.42 (0.84)	7.89 ** (1.94)	9.36 (0.49)
Are these images acceptable for diagnostic use? ^b^					0.97 (0.17)	0.97 (0.17)

Values are means, standard deviations are shown in parentheses. FLAIR = fluid-attenuated inversion recovery, SYN = synthetic scan, CLI = clinical scan. ^a^ Eleven-point scale (0–10); ^b^ Two-point scale (0, 1); * corrected *p* < 0.05; ** corrected *p* < 0.01.

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
