# Peer review of "Reliability of Synthetic Brain MRI for Assessment of Ischemic Stroke with Phantom Validation of a Relaxation Time Determination Method"

_jcm, 2020, doi:10.3390/jcm9061857_

Round 1

Reviewer 1 Report

in file

Author Response

Response to Reviewer 1

Point 1: The phantoms used are very simple in their structure and certainly do not fully describe the examined tissue and relaxation changes. The recorded signal undoubtedly describes changes in free fluid spaces. So we have to deal with a bulk system, and in fact the brain tissues are anisotropic structures that affect relaxation processes. This should be better described and explained. I suggest adding a description and discussion on the impact on research of other types of phantoms: capillary, laminar, which would better reflect reality. It's probably task for the future, but it should be subject to informed discussion. Such examples of phantoms are in, for example, papers [1-6]. Although they were mostly used for DWI / DTI, which is not currently applicable to the synthetic MRI, they contain structures that better reflect reality and can be used as standards for T1 or T2.

Response 1: Thank you for your treasurable suggestion. We now include the simple structure of our phantoms as a limitation. We modified the section to read (page 6, 2nd paragraph): “Our study has two limitations: the simple structure of our phantoms and the small sample size. First, in this study, we designed and used 2 phantoms with 14 specific concentrations of aqueous CuSO4, showing that the T1- and T2-values are similar to those of brain tissue, thus validating the accuracy of quantitative relaxation times acquired by synthetic scanning. However, brain tissues are anisotropic structures and can affect relaxation processes, and the simple structure of our phantoms could not describe actual brain tissue but only changes in free fluid spaces. Some have demonstrated that other types of phantoms could better reflect reality in brain tissue compared to ours [24-29]; therefore, we suggest using those phantoms in further studies validating quantitative relaxation times. Second, this infers that our results are not generalizable to other comparisons between synthetic and clinical scanning in chronic ischemic stroke. Therefore, these results cannot be applied to acute/sub-acute stroke or hemorrhagic stroke. Further investigations using images in acute, sub-acute, and hemorrhagic stroke are now necessary.

Point 2: It should also be explained why the phantoms proposed by the authors can be considered as the first step in this work at this stage. And what the authors propose next.

Response 2: Thank you for your comment. Recently, Tanenbaum et al. revealed that the overall diagnostic quality of synthetic MR images was not inferior to clinical imaging on a 5-point Likert scale, and the legibility of those two scans highly agreed in morphology. However, differences in imaging contrast levels were apparent, particularly in T2-FLAIR images. To provide an overall exploration of synthetic brain scans, we aimed to validate the imaging accuracy in relaxation time, quantifying the reliability of clinical usage. Therefore, our study was conducted to validate the reliabilities of T1 and T2 relaxation times obtained from synthetic MRI scans and to investigate the achievability of contrast between stroke lesion and brain tissue on a synthetic MRI. This is the reason that phantom validating is the first step in our study.

Point 3: Phantoms should also be described numerically, their dimensions, i.e. a simulation of a certain porous (biological) system.

Response 3: Thank you for your kind reminder; we added to the description of Figure 1: “cylindrical structure 7.1 cm2 in cross-sectional area and 6.0 cm in height.”(Page 2, 2nd paragraph)

Reviewer 2 Report

Li and colleagues validated the reliability of relaxation time measurement in synthetic MRI of homemade phantoms and compared the diagnostic suitability of synthetic imaging to that of conventional MRI scans in detecting ischemic lesions. Overall, the study is scientifically-sound and well conducted and the findings are of potential interest for the specialist readers.

I have only two minor points:

  • in paragraph 2.1. Synthetic MRI Technique, insert the appropriate unit of measurment after the number 14 ("...and echo-train length, 14.")
  • please, cite the R software appropriately.

    To cite R in publications, use

    @Manual{, title = {R: A Language and Environment for Statistical Computing}, author = {{R Core Team}}, organization = {R Foundation for Statistical Computing}, address = {Vienna, Austria}, year = YEAR, url = {https://www.R-project.org} }

    where YEAR is the release year of the version of R used and can determined as R.version$year.

    Citation strings (or BibTeX entries) for R and R packages can also be obtained by citation().

Author Response

Response to Reviewer 2

Point1: In paragraph 2.1. Synthetic MRI Technique, insert the appropriate unit of measurement after the number 14 ("...and echo-train length, 14.")

Response 1: Thank you for your kind reminder. We corrected this to read, “echo-train length, 14 echoes.”(Page 2, 4th paragraph)

Point 2: Please, cite the R software appropriately. To cite R in publications, use @Manual{, title = {R: A Language and Environment for Statistical Computing}, author = {{R Core Team}}, organization = {R Foundation for Statistical Computing}, address = {Vienna, Austria}, year = YEAR, url = {https://www.R-project.org} }. where YEAR is the release year of the version of R used and can determined as R.version$year. Citation strings (or BibTeX entries) for R and R packages can also be obtained by citation().

Response 2: Thank you for your kind reminder. We modified the citation to read, “R Core Team 2018. R: a language and environment for statistical computing. R Foundation for Statistical Computing, Vienna, Austria. URL http://www.R-project.org.”(Page 3, last 2 sentences)